# Automatic Evaluation for Bioengineering of Human Artificial Ovary: A Model for Fertility Preservation for Prepubertal Female Patients with a Malignant Tumor

**DOI:** 10.3390/ijms232012419

**Published:** 2022-10-17

**Authors:** Wanxue Wang, Cheng Pei, Evgenia Isachenko, Yang Zhou, Mengying Wang, Gohar Rahimi, Wensheng Liu, Peter Mallmann, Vladimir Isachenko

**Affiliations:** 1Department of Obstetrics and Gynaecology, Medical Faculty, Cologne University, 50931 Cologne, Germany; 2NHC Key Laboratory of Male Reproduction and Genetics, Guangdong Provincial Reproductive Science Institute, Guangdong Provincial Fertility Hospital, Guangzhou 510062, China

**Keywords:** artificial ovary, fertility preservation, prepubertal cancer, primordial follicles, imaris reconstruction

## Abstract

Introduction: The in vitro culture of primordial follicles is the only available option for preserving fertility in prepubertal girls with malignant tumors. The cultivation of primordial follicles in scaffolds as artificial ovaries is a promising approach for this. Methods: Dissociated follicles were placed into an artificial ovarian scaffold composed of fibrinogen and thrombin. The follicles were cultured in a dish dedicated to live cell imaging and observed for growth using immunofluorescence and development via optical microscopy. The morphology of the follicles in the scaffold was three-dimensionally reconstructed using the Imaris software. Growth and development were also quantified. Results: The morphology of artificial ovaries began to degrade over time. Within approximately 7 days, primordial follicles were activated and grew into secondary follicles. A comparison of optical and confocal microscopy results revealed the superior detection of live cells using confocal microscopy. The three-dimensional reconstruction of the confocal microscopy data enabled the automatic enumeration and evaluation of the overall morphology of many follicles. Conclusions: The novel artificial ovary-enabled primordial follicles to enter the growth cycle after activation and grow into secondary follicles. The use of a fibrin scaffold as a carrier preserves the developmental potential of primordial germ cells and is a potentially effective method for preserving fertility in prepubertal girls.

## 1. Introduction

Over the past 40 years, the treatment of childhood cancer has become increasingly successful. Most childhood cancers can now be cured with multidrug chemotherapy combined with surgery and radiation therapy [1,2]. The prognosis of prepubertal cancer has markedly improved, with approximately 75% of patients surviving for more than 5 years [3,4]. However, most childhood cancer patients receive radiotherapy, which is highly destructive to prepubertal reproductive organs and can lead to the permanent loss of fertility [5]. Therefore, maintaining the fertility of prepubertal females treated for cancer is necessary [6].

Fertility preservation options for prepubertal girls with malignant tumors include ovarian transposition, ovarian tissue cryopreservation, auto-transplantation after sexual maturity, and ovarian tissue culture in vitro [7,8]. However, these options are associated with the risk of re-introducing cancer cells [9,10,11]. Ovarian tissue dissociation using enzymes followed by the selection of follicles for in vitro cultures is considered a relatively safe method with a low risk of introducing cancer cells [12,13,14].

The in vitro cultures of ovarian tissue and germ cells, including a range of biological scaffolds and artificial ovaries, is a promising technique for this [14]. The in vitro cultures of primordial germ cells can be performed in the absence or presence of a scaffold. Scaffold-free cultured follicles can be reshaped using special wells with round bottoms [15] or microfluidic systems in which the follicular microenvironment is systematically regulated in real time [16]. Researchers have experimented more extensively with scaffolded cultures. Different formulations of biological scaffolds include alginate, fibrin, a decellularized ovarian extracellular matrices, three-dimensional (3D)-printed ovaries, and others [17]. Although artificial ovaries have not been used for live human births, studies have demonstrated the success of follicular scaffolds in mice, where follicles can be retrieved and healthy offspring can be born via in vitro fertilization and embryo transfer [18,19,20,21].

This study aimed to create a fibrin artificial ovarian scaffold for prepubertal women with cancer as a reliable and reproducible method to evaluate growth and development and for automated enumeration.

## 2. Results

Nuclear and cytoplasmic staining revealed single follicles (Figure 1G–I). The Appendix A of a representative isolated follicle is included for improved visualization. The secondary follicle comprised two layers of granulosa cells and an oocyte displaying more intense cytoplasmic staining.

Dark-field photography using the Zeiss PH0.4 dark-field mode was necessary (Figure 2D–L). Appendix A provides detailed information on the microscope setup. Many follicles transparent and light red or light brown in color, which are hallmarks of viable follicles, were observed (Figure 2D–F,I,K,L).

Table 1 presents the results of the quantitative analysis of follicles in artificial ovaries using optical microscopy. From days 1 to 3, 55 primordial follicles were activated and formed primary follicles. Their diameters exceeded 60 µm. Six primary follicles developed into secondary follicles. Optical microscopy revealed the number of follicles and their diameters. The technique was not effective in discerning whether the follicles were alive, dying, or dead. For these determinations, a cell viability marker was necessary, described as follows. On day 5, more primordial follicles developed into primary follicles, and more secondary follicles were evident. On day 7, 18 primordial follicles became activated and 11 more primary follicles displayed extensions of the granulosa cell layers to become secondary follicles.

The RedDot fluorescence marker was used for live cells (Figure 3A,B). Viable cells were identified using markers upon image denoising (Figure 3B). The quantitative results of the Imaris reconstruction indicated the presence of markedly fewer follicles on day 7 than on day 5 (20.875% fewer). This finding identified the presence of dead follicles after 1 week of culture that were not discerned using optimal microscopy. Use of the change in the trend of the overall surface area of the spheroid follicles (Figure 3C,D) to reflect the overall status of follicle growth might generate results that more closely reflect the actual situation. Approximately 20% of the follicles died by day 7.

## 3. Discussion

An artificial ovary forms a gaping mesh structure that provides support to the follicles. The structural microenvironment that is maintained mimics the surrounding follicles in human ovarian tissues [22]. The elasticity of the fibrin meshwork also provides extracellular matrix-like stiffness to support follicle growth and development [23]. The mesh structure further allows for the rapid and complete penetration of various nutrients, which can provide different hormones and nutrients for each stage of follicular growth and maintain the stability of the intrafollicular microenvironment [22,23,24,25].

The ultimate goal of an artificial ovary is re-transplantation into the human body. Therefore, its components must be biologically safe and tolerable. The scaffold should be degradable for follicle growth and migration, as folliculogenesis involves a change in diameter from 19 to 30 μm in the primordial stage and 100 to 110 μm in the mature stage. Scaffolds must also be resistant to human body temperature [26] and allow signals to pass to the cells and microenvironment. The 3D matrix should be highly permeable to allow for the diffusion of nutrients and signals in and out of the scaffold. Isolated follicles are fragile but stable when embedded in 3D stents and are easy and safe to manipulate without disrupting the follicular structure [27]. Tissue engineering using biomaterials supporting artificial ovaries, ranging from natural (collagen, plasma clots, alginate, fibrin, decellularized tissue, and others) to synthetic (polyethylene glycol, 3D-printed ovaries, and others) polymers, has yielded promising results in animal research models [28,29,30,31]. Natural polymers are not rigid enough to support mechanical structures. However, they also offer advantages such as good cell adhesion, migration, and signal communication. Synthetic polymers have excellent mechanical properties that allow them to act as supports when transplanted into the human body [32].

We used only fibrin, instead of alginate, to demonstrate an artificial ovary matrix. This is because fibrin is a natural component regularly present in the human body [33,34]. In clinical practice, dermatologists usually use it to promote wound healing [35,36]. However, alginate originates from plants, and is an artificial component [37,38]. Moreover, this study represents preliminary research, before the clinical use of an artificial ovary. We aimed to explore the follicle changes in the fibrin matrix after in vitro culture and as a pre-clinical study. In terms of its prospects for clinical application, fibrin has more advantages than alginate.

The artificial ovary model is novel in several respects. Fibrin scaffolds were fabricated as a result of previous investigations [39,40,41,42]. However, the scaffolds were devised differently, as they facilitated the determination of counts using Imaris software. Simultaneously, the special flat shape of the fibrin scaffold, and the use of suitable culture dishes (Figure 3B), photography, and data acquisition allowed for tile scanning using inverted confocal microscopy. Automatic counting is highly efficient and will be valuable for the subsequent establishment of a multi-sample culture system. The fibrin scaffold that we used supported follicle growth and development. The study findings highlight the potential value of fibrin for future clinical applications as an artificial ovarian scaffold in human pelvic and abdominal cavities. Fibrin is a Food and Drug Administration-approved material for wound treatment. Accordingly, its safety and compatibility with humans have been well-established [43,44,45,46].

This preclinical study is important for prepubertal girls with aggressive tumor metastases, such as those with leukemia and lymphoma, which are two important pediatric cancers [47,48,49]. The main target function of the artificial ovary is to prevent the re-implantation of malignant cells and mimic the function of the ovary. The approach described in the present study could offer prepubertal girls the opportunity to conceive and recover endocrine functions without cyclic hormone replacement therapy.

Ensuring that the follicle isolation process is free from malignant cell transfer is critical for artificial ovaries as both follicles and malignant cells enter the digestive fluid. Follicles can also be contaminated with malignant cells during the follicle selection process, and these could be reimplanted into the artificial ovaries. Soares et al. [50] grafted 100 leukemic cells into artificial ovaries and transplanted them into mice. Leukemia was not evident after 20 weeks, and immunohistochemistry and polymerase chain reactions results were negative using the recovered ovaries. The authors concluded that grafting 100 leukemic cells was not sufficient to induce leukemia [50]. Another study verified that malignant cells could be effectively eliminated by washing the follicles three times without affecting their viability [51]. These results were confirmed by performing repeated experiments using multicolor flow cytometry [52]. At present, it remains unclear how many malignant cells must be implanted to cause cancer recurrence. There is no evidence that the load of recurrence after implantation varies according to the tumor type. Therefore, future studies should address these issues.

The constant level of estradiol in the culture medium after cultivation can be explained by the large volume of this culture medium (4–5 mL) compared with the small volume of all follicles (2 × 10^−6^ mL). The amount of estradiol 17-β involved in the metabolism of follicles was below the physical sensitivity threshold of the device.

## 4. Materials and Methods

The process used for the artificial ovary process is detailed in Figure 4.

### 4.1. Ethics Permission and Informed Consent Statement

This study was approved by the Ethics Boards of the University of Cologne (applications 999,184 and 13−147). Written informed consent was obtained from all participating patients who underwent laparoscopic surgery at the Department of Gynecology and Obstetrics of the University Hospital of Cologne.

### 4.2. Cryopreservation of Ovarian Tissue and Thawing

All acquired ovarian tissues were cryopreserved and stored in the cryobank of the Maternal Hospital of Cologne University. According to the protocol approved by our department, 10% of ovarian tissues collected from patients was used for “patient-oriented research” to assess tissue viability for re-transplantation. The cryopreservation of ovarian cortex pieces (OCPs) was performed according to our previously published protocol, with a 24h pre-cooling time added before freezing [53,54]. On the day of freezing, the OCPs were placed in basal medium for 30 min at room temperature. After this, the OCPs were placed into standard 5 mL cryovials (Thermo Fisher Scientific, Waltham, MA, USA) and frozen in an IceCube 14S programmable freezer (Sy-Lab, Neupurkersdorf, Austria). For thawing, the cryovial was held at room temperature in the air for 30 s before direct immersion in a boiling water bath (100 °C) for 60 s. The thawed OCPs were immediately transferred from the cryovials to 10 mL of thawing solution (basal medium containing 0.5 M sucrose). After the stepwise dilution of the cryoprotectants, the OCPs were transferred to the basal medium for 10 min.

### 4.3. Isolation of Follicles

An enzyme solution was prepared prior to cutting the tissue fragments. Two milliliters of enzyme solution was needed for every 100 mg of tissue. The solution contained Liberase DH (0.28 Wünsch units/mL), DNase I (10 µg/mL), and Dulbecco’s phosphate-buffered saline containing Mg^2+^ and Ca^2+^ [55]. Two No. 22 scalpels, one held in each hand, were used to cut the ovarian tissue fragments in a Petri dish with a 5 cm diameter on ice. Fragments were cut as quickly as possible until they formed a mash. The mashed tissue was transferred to a centrifuge tube using a Pasteur pipette, and the prepared enzyme solution was added. The mashed tissue was dispersed in the enzyme solution and shaken for 60 min at 130 rpm in a 37 °C incubator, with mixing of the suspension every 15 min using a 1000 µL pipette. Neutral red solution (working concentration 50 µg/mL) was added for 15 min before the end of dissociation. Enzymatic digestion was inhibited via the addition of an equal volume of L-15 medium containing 20% fetal calf serum at 4 °C. Isolated follicles were collected with 125 μm V-denuded capillaries (Vitromed GmbH, Jena, Germany) on a COOK micropipette using a stereo microscope (Nikon, Tokyo, Japan). Oblique coherent contrast was used to obtain the best three-dimensional perspective and to recognize follicles more effectively. Follicle diameter was measured based on the basement membrane surrounding the granulosa cell layer(s), which served as the follicular boundary [56,57]. The follicle stage was classified as previously described [26,27] Primordial follicles (<60 μm) comprised oocytes surrounded by a single layer of flattened pre-granulosa cells. Primary follicles (>60 to ≤75 μm) comprised oocytes with a single layer of cuboidal granulosa cells. Secondary follicles (>75 to <200 μm) comprised at least two complete layers of granulosa cells. The collected follicles were washed four times with pre-cooled basal medium to remove stromal cells. The washed follicles were transferred to an Eppendorf tube and aspirated as much as possible by pipetting. This enrichment step minimized the total amount of fluid in the tube as the subsequent fibrin gels could be diluted.

### 4.4. Follicles Encapsulation and In Vitro Culture

TISSEEL Fibrin Sealant (Baxter International Inc., Deerfield, IL, USA) was used to encapsulate the isolated follicles. Fibrinogen and thrombin at final concentrations of 45.5 mg/mL and 10 IU/mL, respectively, are desirable for optimal encapsulation. Both components were quickly mixed in an Eppendorf tube using a pipette and vortexed. The follicle solution (30–50 follicles for every artificial ovary) was then added dropwise (30 µL) to form a nearly gelatinous mixture. The mixture was then dropped into a Petri dish designed for live cell imaging (WillCo Wells B.V, Amsterdam, The Netherlands). Follicles were cultivated in alpha-modified Eagle’s minimum essential medium (α-MEM, Gibco BRL; Life Technologies, Carlsbad, CA, USA) supplemented with 15% fetal calf serum, 2 mmol L-glutamine (Gibco BRL), insulin-transferrin-sodium selenite supplement (ITSE; Sigma-Aldrich, St. Louis, MO, USA), ascorbic acid (50 µg/mL), 100 IU/mL penicillin, 0.1 mg/mL streptomycin, and 300 mIU/mL human recombinant follicle stimulation hormone (Gonal F^®^; Serono Pharma GmbH, Munich, Germany) in a humidified incubator in a 5% CO_2_ atmosphere at 37 °C [55]. Half the volume of the medium was changed every 48 h and all the artificial ovary samples were cultured for 7 days.

### 4.5. Artificial Ovary Imaging

Optimal microscopy of the neutral red-stained tissue was used to evaluate follicle morphology. Details of the 3D imaging in the Ph0.4 mode are provided in Appendix A. The shape differences and layer differences could be distinguished with bright field images using Ph0.4 mode on Zeiss Anxiovert 40 CFL microscopes. this is a phase contrast setting in a bright field, and all the cells shapes and follicle sizes can be distinguished (Appendix A). In addition, under the stereoscope (Nikon SMZ18) in OCC mode (see the detailed parameters in Appendix A), it was possible to distinguish cell shapes and follicle layers.

Follicles were stained with RedDot™ (Biotium, Biotechnology company in Fremont, CA, USA), as shown in Figure 3 and Appendix A, which is a fluorescence marker with λEx/λEm = 662/694 nm (with DNA), for detection in the Cy5 channel. RedDot™1 is a far-red cell membrane-permeant nuclear dye similar to Draq5™. This dye is ideal for specifically staining the nuclei of live cells. Confocal microscopy was performed using an LSM 710 microscope (Carl Zeiss, Jena, Germany) with a 3D imaging setup comprising tiles scanning and a transmission depth from 300–500 µm. Photographs were taken at 4 µm intervals. The original microscopy data were analyzed using Imaris 9.0 software. Imaris imaging reconstruction and analysis were performed using the “Surpass” and “3D view” modes. The threshold was changed to improve the accuracy of follicle identification. RedDot fluorescence staining was used to detect the live cells.

## 5. Conclusions

The artificial ovary, constructed as described in the present study allows primordial follicles to enter the growth cycle after being activated and supports their further growth into secondary follicles. The fibrin scaffold is a potentially valuable means to protect the fertility of prepubertal girls with malignant tumors by preserving the developmental potential of the primordial germ cells.

## Figures and Tables

**Figure 1 ijms-23-12419-f001:**
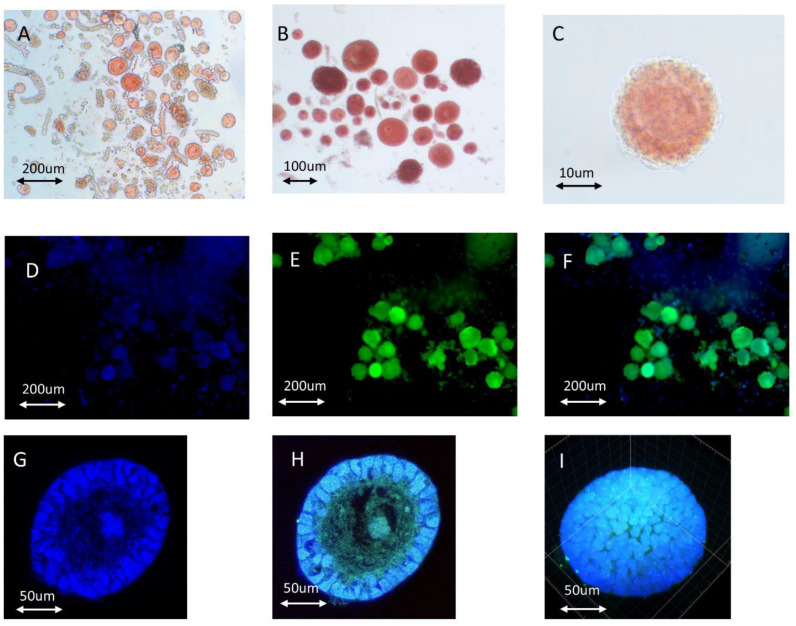
Vitality and morphology evaluations of isolated follicles. (**A**) Freshly isolated follicles in an enzyme solution before manipulated enrichment using a capillary. The follicles were stained with neutral red. (**B**) Follicles manipulated and enriched using a 125 µm capillary in MOPS solution. (**C**) Optical microscopy image of isolated primordial follicles stained with neutral red. (**D**) Hoechst-stained follicles observed via fluorescence microscopy. (**E**) Calcein-AM-stained follicles after isolation observed via fluorescence microscopy. (**F**) Merged image of the images shown in panels (**D**,**E**). (**G**) Hoechst staining of an isolated secondary follicle. (**H**) Merged confocal photography of Hoechst and Calcein-AM-stained secondary follicle. (**I**) Spherical screenshot of the isolated secondary follicle.

**Figure 2 ijms-23-12419-f002:**
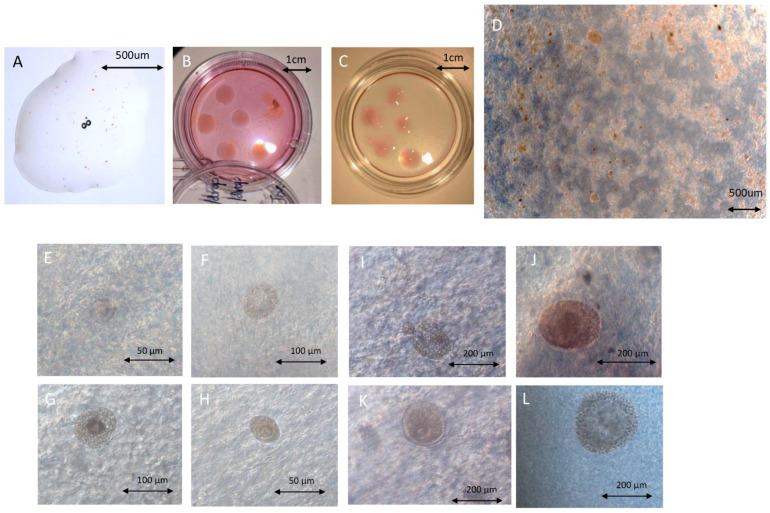
Photography of follicles in fibrin scaffolds. (**A**) Fibrinogen–thrombin complex including follicles in the unsolidified state. (**B**,**C**) General view of an artificial ovary on day 0 (**B**) and day 7 with a dissociated scaffold (**C**). (**D**) Optimal microscopy image of artificial ovary on day 0. (**E**,**F**) Viable primordial follicle in fibrin on day 0 (**E**) and day 3 (**F**). (**G**) Dead primary follicle on day 7. (**H**) Viable primordial follicle on day 7. (**I**) Viable secondary and primordial follicles in fibrin on day 3. (**J**) Dying secondary follicle in fibrin on day 7. (**K**) Viable secondary follicle in fibrin on day 3. (**L**) Viable secondary follicle in fibrin on day 7.

**Figure 3 ijms-23-12419-f003:**
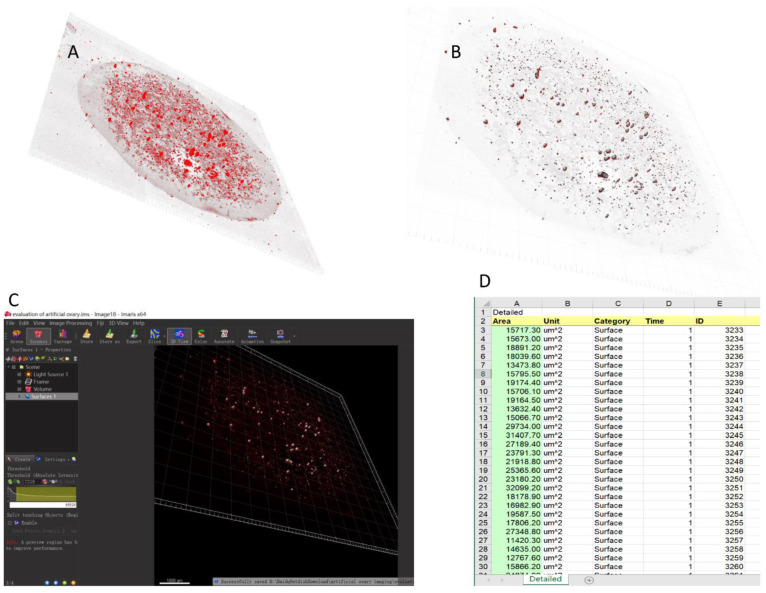
Automatic calculation and analysis system for the human artificial ovary. (**A**) Three-dimensional reconstruction of artificial ovaries imaged using Imaris 9.0 software with RedDot staining. (**B**) Distribution of follicles for analytical counting after image denoising. (**C**) Interface and general parameter settings of automatic counting and analysis system. (**D**) After automatic counting, the system generated a data file in txt format.

**Figure 4 ijms-23-12419-f004:**
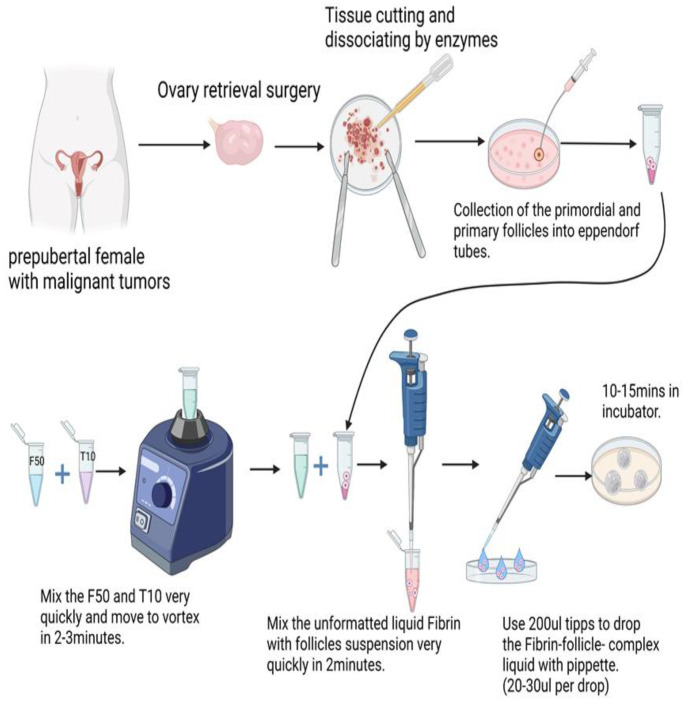
Construction of the artificial ovary from tissue to scaffold.

**Table 1 ijms-23-12419-t001:** Quantitative analysis of follicles on the artificial ovary using optical microscopy vs. reconstruction imaging with confocal microscopy.

	Optical Microscopy (*n* = 5) (Follicles Amounts, Total = 212) ^1^	Confocal Microscopy + Imaris Reconstruction (*n* = 5) (Follicle Spherical Surface Area, μm^2^) ^1^
Day 1	Primordial = 169 Primary = 26 Secondary = 17	2,365,971.01
Day 3	Primordial = 104 Primary = 85 Secondary = 23	2,629,148.77 (+11.123%)
Day 5	Primordial = 39 Primary = 128 Secondary = 45	3,040,046.53 (+15.629%)
Day 7	Primordial = 21 Primary = 135 Secondary = 56	2,405,424.01 (−20.875%)

^1^*n* = drops of the artificial ovary in one culture dish; normally every drop included 30–50 follicles.

## Data Availability

The data used to support the findings of this study are included in the article.

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
