# Peer review of "Automatic Evaluation for Bioengineering of Human Artificial Ovary: A Model for Fertility Preservation for Prepubertal Female Patients with a Malignant Tumor"

_ijms, 2022, doi:10.3390/ijms232012419_

Round 1

Reviewer 1 Report

The idea of fertility preservation is novel and the authors have put some effort into culture of ovarian follicles, however from the present data the outcomes are not convincing.

1. The methods and the results observed from the in vitro culture of ovarian follicles are not well described nor well presented.

2. It's surprising how the authors distinguished primary and secondary follicles from the bright field images. 3. As per the authors the criteria of transparent/ light red/ light brown to select live or dead follicles is not convincing. 4. The authors should stain follicles with markers to distinguish them. If the follicles are dead by day 7, then how the idea of fertility preservation is postulated here?   5.The supplementary figure 1 is missing and the authors should explain the supplementary videos. Both videos look the same. 6. Calcein AM staining should be done along with EtBr to mark live and dead cells.

Reviewer 2 Report

In the current study the authors designed an artificial ovarian scaffold made with fibrinogen and thrombin where isolated primordial follicles were cultured and growed into secondary follicles. The results obtained are of primary importance to elaborate new strategy for perfrtility preservation in oncological patients, especially pre-pubertal girls. The manuscript is clear and well written. However, I would suggest to further clarify how the madulla microenvironment, of primary importance for follicular growth, was replaced in this model and how molecular, hormonal and vascular milieu can be garanteed su support follicular transition.

Round 2

Reviewer 1 Report

.